# Krüppel-like Factor 5 Plays an Important Role in the Pathogenesis of Chronic Pancreatitis

**DOI:** 10.3390/cancers15225427

**Published:** 2023-11-15

**Authors:** Maryam Alavi, Ana Mejia-Bautista, Meiyi Tang, Jela Bandovic, Avi Z. Rosenberg, Agnieszka B. Bialkowska

**Affiliations:** 1Department of Medicine, Renaissance School of Medicine, Stony Brook University, Stony Brook, NY 11794, USAmeiyi.tang@stonybrookmedicine.edu (M.T.); 2Department of Pathology, Renaissance School of Medicine, Stony Brook University, Stony Brook, NY 11794, USA; 3Department of Pathology, Johns Hopkins University, Baltimore, MD 21217, USA; arosen34@jhmi.edu

**Keywords:** Krüppel-like factor 5, Kirsten rat sarcoma viral oncogene homolog, pancreatic intraepithelial neoplasia, acinar-to-ductal metaplasia

## Abstract

**Simple Summary:**

This report shows that KLF5 plays a crucial role in the progression of chronic pancreatitis to pre-neoplastic precursor lesions. Notably, the deletion of *Klf5* from pancreatic acinar cells by itself or in the context of activating KRAS mutation resulted in a significant reduction in inflammatory and stromal activation and, thus, inhibition of chronic pancreatitis progression. Furthermore, this study demonstrated that KLF5 exerts its role by directly activating profibrotic and inflammatory cytokines. Using ChIP-PCR, we demonstrated that KLF5 binds directly to the *Il1b*, *Il6*, and *Tgfb1* gene promoters in vitro and in vivo. Deletion of *Klf5* during chronic pancreatitis development and progression resulted in a decrease of several vital chemokines, such as CCL6, CCL11, and CCL21. In summary, we showed that KLF5 regulates inflammatory and fibrotic responses during chronic pancreatitis.

**Abstract:**

Chronic pancreatitis results in the formation of pancreatic intraepithelial neoplasia (PanIN) and poses a risk of developing pancreatic cancer. Our previous study demonstrated that Krüppel-like factor 5 (KLF5) is necessary for forming acinar-to-ductal metaplasia (ADM) in acute pancreatitis. Here, we investigated the role of KLF5 in response to chronic injury in the pancreas. Human tissues originating from chronic pancreatitis patients showed increased levels of epithelial KLF5. An inducible genetic model combining the deletion of *Klf5* and the activation of *Kras^G12D^* mutant expression in pancreatic acinar cells together with chemically induced chronic pancreatitis was used. The chronic injury resulted in increased levels of KLF5 in both control and *Kras^G12D^* mutant mice. Furthermore, it led to numerous ADM and PanIN lesions and extensive fibrosis in the KRAS mutant mice. In contrast, pancreata with *Klf5* loss (with or without *Kras^G12D^*) failed to develop ADM, PanIN, or significant fibrosis. Furthermore, the deletion of *Klf5* reduced the expression level of cytokines and fibrotic components such as *Il1b*, *Il6*, *Tnf*, *Tgfb1*, *Timp1*, and *Mmp9*. Notably, using ChIP-PCR, we showed that KLF5 binds directly to the promoters of *Il1b*, *Il6*, and *Tgfb1* genes. In summary, the inactivation of *Klf5* inhibits ADM and PanIN formation and the development of pancreatic fibrosis.

## 1. Introduction

Pancreatitis (acute and chronic) is a major gastrointestinal disorder, with acute pancreatitis being one of the most frequent causes of gastrointestinal-related hospitalization [1,2]. Acute pancreatitis is caused by acute injury to acinar cells, characterized by aberrant enzyme synthesis in the acinar cells, followed by the inhibition of enzyme secretion, increased inflammation, and transient damage to the tissue. Chronic injury, with continuous inflammation, increased activity of stellate cells, fibrosis, and atrophy of acinar cells, can progress to pancreatic intraepithelial neoplasia, a precursor to pancreatic cancer [3,4,5,6,7,8,9,10]. During chronic pancreatitis, recurrent injury to the pancreatic acinar cells due to premature activation of pancreatic enzymes results in the autodigestion of pancreatic parenchyma. Subsequently, proinflammatory mediators promote immune cell infiltration [11,12]. In addition, injured pancreatic acinar cells and immune cells produce cytokines and growth factors such as IL-1, IL-6, IL-8, TGF-β, TNF-α, PDGF, and VEGF that simulate pancreatic stellate cells (PSCs). In turn, activated PSCs produce autocrine factors (mainly PDGF, TGF-β, IL-1, IL-6, and Endothelin-1) that support their active profibrotic state. These factors mediate extensive extracellular matrix remodeling, including the deposition of collagen fibers with progressive fibrosis and stiffening of the tissue, creating an environment supportive of the evolution of neoplasia [13,14,15,16,17,18,19,20].

KRAS is mutated in more than 90% of pancreatic cancers [21]. It is well-established that progression from the normal pancreas to pancreatic cancer is initiated by a point mutation in the *KRAS* gene, followed by mutation(s) in tumor suppressor genes such as *CDKN2A*, *TP53*, and *DPC4* [22,23,24,25]. Importantly, mutations in KRAS are often found in early/pre-neoplastic lesions resulting from chronic pancreatitis [26,27]. Elevated KRAS levels in pancreatic acinar cells with a persistent inflammatory injury during chronic pancreatitis lead to fibrosis and early neoplasia and increase the likelihood of developing pancreatic cancer [28,29,30,31].

Krüppel-like factor 5 (KLF5) is a member of the specificity proteins and Krüppel-like factor (KLF) family of transcription factors that are characterized by a triple-zinc finger DNA-binding domain at the carboxyl termini [32]. KLF5 can promote proliferation, embryonic stem cell self-renewal, cell survival, and differentiation [33,34,35,36,37,38,39]. Notably, KLF5 plays an essential role in the physiology and pathophysiology of extensive regions in the digestive tract, including the oral mucosa, esophagus, liver, stomach, and small and large intestines [40]. KLF5 mediates the activity of pathways which are central to inflammation and/or carcinogenesis, such as MAPK, PI3K, TCF/β-catenin, and NFkB, and, conversely, is regulated by these pathways [36,41,42,43,44,45,46]. It is an essential mediator of tumor development and progression [47].

KLF5 is overexpressed in pancreatic ductal adenocarcinomas (PDACs) compared to normal tissue. It was identified as a pro-oncogenic factor using an RNA interference-based functional screen in pancreatic cancer cell lines [48,49]. Importantly, data generated by the TCGA research network showed a strong negative correlation between KLF5 positivity and patient survival [50]. Our previous study showed that KLF5 is expressed in most human PDACs and the mouse model of oncogenic KRAS-induced PanIN formation [51]. Furthermore, we showed that inhibition of *Klf5* reduced the proliferation of pancreatic cancer cells in vitro and reduced tumor growth in a xenograft model of pancreatic cancer [51]. We observed that KLF5 is not expressed in normal pancreatic acinar cells, but its levels increased in foci undergoing ADM or PanIN transformation. We demonstrated that KLF5 is necessary and sufficient for a step-wise progression from acinar cells to ADM to PanIN during acute injury to the pancreas independently of KRAS mutation [51].

Here, we demonstrate the role of KLF5 in chronic injury and its role in regulating the pancreatic microenvironment during chronic pancreatitis development. To this end, we employed an animal model with inducible deletion of *Klf5* or activation of *Kras^G12D^* mutant in pancreatic acinar cells combined with repeated injection of cerulein to mimic chronic pancreatitis. Our results showed that the deletion of *Klf5* (with or without *Kras^G12D^*) from pancreatic acinar cells inhibits the development of chronic injury-mediated ADM and PanIN. In addition, the levels of DNA damage and cell proliferation were significantly reduced without *Klf5*. Notably, qPCR data showed that the deletion of *Klf5* from pancreatic acinar cells reduced the expression levels of multiple proinflammatory and profibrotic factors. Furthermore, ChIP-PCR demonstrated that KLF5 directly binds to the promoters of *Il1b*, *Il6*, and *Tgfb1*. In summary, we showed that KLF5 plays a vital role in mediating ADM and PanIN of chronic pancreatitis by regulating the inflammatory response of acinar cells.

## 2. Materials and Methods

### 2.1. Animal Studies

All mice were housed in the Division of Laboratory Animal Resources (DLAR) at Stony Brook University. All studies involving mice have been approved by the Stony Brook University Institutional Animal Care and Use Committee (IACUC) and maintained on a 12:12 h light-dark cycle. The DLAR facility has optimized conditions, in terms of well-regulated temperature, humidity, and light settings, to ensure a stable, reproducible environment for animal growth. Mice with the following genotypes were used in this study: *Ptf1aCre^ERTM^*;*Rosa26^tdTomato/+^*, *Ptf1aCre^ERTM^*;*Rosa26^tdTomato/+^*;*Kras^G12D^*, *Ptf1aCre^ERTM^*;*Rosa26^tdTomato/+^*;*Klf5^fl/fl^*, and *Ptf1aCre^ERTM^*;*Rosa26^tdTomato/+^*;*Kras^G12D^*;*Klf5^fl/fl^*. All mice were maintained on a mixed background. We employed 8- to 12-week-old and gender-matched mice in this study. To induce Cre-mediated recombination, we performed three intraperitoneal injections of tamoxifen dissolved in corn oil every other day in week 1 (Sigma-Aldrich, St. Louis, MD, USA, T5648; 3 mg/injection). Corn oil-injected mice were used as controls. Chronic pancreatitis was induced one week after the first tamoxifen injection using I.P. injections of 100 µL cerulein (BACHEM, Bubendorf, Switzerland, H3220) at 50 µg/kg of body weight in DPBS (Fisher Scientific, Hampton, NH, USA, 21-031-CV) six times a day, three times a week, for four weeks. DPBS-injected mice were used as controls. Pancreatic tissues were collected three days after the last cerulein or DPBS injection. Blood samples were collected three days after the last injection via superficial temporal vein sampling in a Micro sample tube Serum Gel (Sarstedt, Nümbrecht, Germany, 41.1378.005). The tubes were centrifuged at 10,000× *g* for 10 min at room temperature, kept at 4 °C, and shipped for analysis.

### 2.2. Histology

Pancreatic tissues were fixed overnight in 10% neutral buffered formalin (Azer Scientific, Morgantown, PA, USA, CUNBF-5G), transferred into 70% ethanol, processed using an automated processor, and paraffin-embedded in the Histology Research Core at Stony Brook Cancer Center. Five-micron thick histologic sections of the pancreas were used for all stains. Images were captured using a Nikon Eclipse 90i microscope (Nikon, Melville, NY, USA).

#### 2.2.1. Hematoxylin and Eosin Stain (H&E)

FFPE slides were de-paraffinized in xylene and rehydrated in a decreasing series of ethanol baths (100%, 95%, 70%). Then, they were stained with Hematoxylin Gill solution (Sigma-Aldrich, St. Louis, MD, USA, GHS332-1L) and Eosin Y solution (Sigma-Aldrich, St. Louis, MD, USA, HT110216). Sections were dehydrated in a series of ethanol baths (70%, 95%, 100%), cleared in xylene, and mounted with Richard-Allan Scientific^®^ Cytoseal™ X.Y.L. Mounting Medium (Fisher Scientific, Hampton, NH, USA, 8312-4). The tissues were assessed by Dr. Jela Bandovic (Department of Pathology, Renaissance School of Medicine at Stony Brook University) using previously established criteria [52,53].

#### 2.2.2. Alcian Blue Stain

The stain was performed according to the manufacturer’s protocol (Vector Laboratories, Newark, CA, USA, H-3501). Briefly, FFPE pancreas sections were de-paraffinized, then hydrated in deionized water, immersed in acetic acid, washed, and immersed in Alcian blue solution for 15 min at 37 °C in acetic acid. After rinsing with distilled water, the slides were counterstained in Nuclear Fast Red, dehydrated, cleared in xylene, and mounted with Richard-Allan Scientific^®^ Cytoseal™ X.Y.L. Mounting Medium (Fisher Scientific, Hampton, NH, USA, 8312-4).

#### 2.2.3. Immunohistochemistry and Immunofluorescence

FFPE slides were de-paraffinized in xylene, rehydrated in a series of ethanol baths (100%, 95%, 70%), and stained as described previously [51]. The slides were incubated with primary antibodies overnight at 4 °C with a gentle shake. Primary antibodies: anti-KLF5 (R&D Systems, Minneapolis, MN, USA, AF3758), anti-KRT19 (Developmental Studies Hybridoma Bank, Iowa City, IA, USA, TROMAIII), anti-Amylase (Santa Cruz Biotechnology, Dallas, TX, USA, sc-46657), anti-α-SMA (Abcam, Cambridge, United Kingdom, ab124964). The secondary antibodies were applied for 1 h at room temperature. Betazoid DAB (Biocare Medical, Bronx, NY, USA, Cat. BDB2004L) was used for immunohistochemical detection. The slides were dehydrated, cleared in xylene, and mounted with Richard-Allan Scientific^®^ Cytoseal™ X.Y.L. Mounting Medium (Fisher Scientific, Hampton, NH, USA, 8312-4). The slides for immunofluorescent detection were counterstained with Hoechst (ThermoFisher Scientific, Bohemia, NY, USA, H3569) and covered with Fluoromount™ Aqueous Mounting Medium (Sigma-Aldrich, St. Louis, MD, USA, F4680).

#### 2.2.4. Fibrosis Assessment

Masson’s Trichrome staining was performed at the Research Histology Core Laboratory in the Department of Pathology at Stony Brook University. The pancreatic sections were stained with a Picro Sirius Red Stain Kit (Abcam, Cambridge, United Kingdom, ab150681) according to the manufacturer’s instructions. The staining, polarized imaging, and analysis were performed in the laboratory of Dr. Rosenberg (Department of Pathology, Johns Hopkins University).

### 2.3. TMA Analysis

Human tissue microarrays BBS14011 and BIC14011a containing de-identified human PDAC tumor samples were purchased from US Biomax, Inc. (Derwood, MD, USA). Human normal-pancreatic tissues and human PDAC tissues were obtained from Stony Brook Medicine Biobank (Stony Brook, NY, USA). T.M.A. arrays and tissues obtained from Stony Brook Medicine Biobank were stained with antibodies against KLF5 (Section 2.2.3).

### 2.4. Isolation of tdTomato-Positive Cells

Pancreata were collected and dissociated with collagenase D (Sigma-Aldrich, St. Louis, MD, USA, 11088858001) in HBSS (Fisher Scientific, Hampton, NH, USA, 14170161) and incubated at 37 °C with shaking. HBSS with 5% FBS (Peak Serum, Inc., Wellington, CO, USA, PS-FB3) was used to stop the reaction, and the cell suspension was passed through a 500 mm nylon mesh following a 100 mm cell strainer. After centrifugation, the cell pellet was re-suspended in 1 mL of DPBS (Fisher Scientific, Hampton, NH, USA, 21-031-CV) with 2% FBS. The cell sorting was based on a tdTomato-positive signal at the Stony Brook Research Flow Cytometry Laboratory facility using a FACSAria IIIu Cell Sorter (B.D. Biosciences, Franklin Lakes, NJ, USA).

### 2.5. Gene Expression Analysis by Quantitative RT-PCR and qPCR Array

According to the manufacturer’s instructions, RNA was extracted using a PureLink RNA Mini Kit (ThermoFisher Scientific, Bohemia, NY, USA, 12183018A). Primers against mouse *Il1b* (Mm00434228_m1), *Il6* (Mm00446190_m1), *Tgfb1* (Mm01178820_m1), *Tnf* (Mm00443258_m1), *Mmp9* (Mm00442991_m1), *Timp1* (Mm01341361_m1), *Col1a1* (Mm00801666_g1), Acta2 (Mm00725412_s1), and *Hprt1* (Mm03024075_m1) were purchased from ThermoFisher Scientific (Bohemia, NY, USA,). cDNA was prepared using a SuperScript^®^ VILO™ cDNA Synthesis Kit (ThermoFisher Scientific, Bohemia, NY, USA, 11754-050). Analysis was performed with Applied Biosystems TaqMan™ Gene Expression Master Mix (ThermoFisher Scientific, Bohemia, NY, USA, 43-690-16) using QuantStudio3 (ThermoFisher Scientific, Bohemia, NY, USA) per standard protocols. Observed CT values were then used to calculate fold change using the 2^−ΔΔCt^ relative quantification method. Mouse *Hprt1* was used as the housekeeping control gene.

### 2.6. Cell Lines, Reagents, and Cell Culture

HEK 293T cell line was purchased from the American Type Culture Collection (ATCC) and cultured according to ATCC instructions. Plasmids encoding *Il1b* promoter fused to mCherry (MPRM39757-PM02), *Il6* promoter fused to mCherry (MPRM38977-PM02), and *Tgfb1* promoter linked to mCherry (MPRM41918-PM02) were purchased from GeneCopoeia (Rockville, MD). EGFP-N1 is a plasmid that originated from Takara-Bio (San Jose, CA, USA), and the plasmid pcDNA3.1HA-KLF5 construct was generated in our laboratory. HEK 293T cells were transfected with pEGFP-N1 and each promoter clone using Lipofectamine 2000 Transfection Reagent (ThermoFisher Scientific, Bohemia, NY, USA, 11668027) according to the manufacturer’s protocol. EGFP and mCherry signals were read using a microplate reader (Spectramax3 Molecular Devices LLC, San Jose, CA, USA) with SoftMax Pro software (version 7).

### 2.7. Chromatin Immunoprecipitation In Vitro and In Vivo

ChIP-PCR was performed using SimpleChIP Enzymatic Chromatin IP Kit (Cell Signaling, Danvers, MA, USA, Cat. # 9003). The KLF5 binding sites were predicted using JASPAR software (version 2022) [54]. For in vitro, HEK 293T cells transfected with plasmids encoding *Il1b* promoter linked to mCherry (MPRM39757-PM02), *Il6* promoter linked to mCherry (MPRM38977-PM02), and *Tgfb1* promoter linked to mCherry (MPRM41918-PM02) were fixed with formaldehyde, and DNA was digested with Micrococcal nuclease. For in vivo, we used frozen pancreas tissues and a modified protocol [55]. Briefly, snap-frozen tissues were put on a glass slide with 1% FA in PBS (40 μL), finely minced, and after 15 min, transferred to a new tube, and incubated in the total volume of 1 mL of 1% FA in PBS at room temperature for 10 min. The reaction was stopped with the addition of glycine and incubation at room temperature for 10 min. The tissues were washed three times using PBS with Protease and Phosphatase inhibitors. The tissues were suspended in ChIP buffer A (Cell Signaling, Danvers, MA, USA, Cat. # 9003), disrupted with tips by going up and down 50 times, and incubated on ice for 10 min. After this step, the manufacturer’s protocol was followed. Digested protein-DNA was incubated with an anti-KLF5 antibody (Abcam, Cambridge, United Kingdom, ab137676) and precipitated using Protein G-coated magnetic beads. Rabbit IgG and Anti-Histone 3 antibodies were used as negative and positive controls, respectively. PCR was run using primer sets specific to the potential binding site: *Il1b* primer set 1 (forward primer: CCTTGACTTCCAGGGATTAGAAA and reverse primer: GCAGAAGTGAAGAGCTGTGA), *Il1b* primers set 2 (forward primer: TCAGGGTAGCAATAGCCTCT and reverse primer: CCTTGGGTTAACTGATTTCACAAC, *Il6* primer set 1 (forward primer: AATAAGGTTTCCAATCAGCCCC and reverse primer: ACAGACATCCCCAGTCTCATA), *Tgfb1* primer set 1 (forward primer: CCACGCTAAGATGAAGACAGTG and reverse primer: CCTGGCTGTCTGGAGGAT), and *Tgfb1* primer set (forward primer: GTTGGTCACCGGCTTTAGTAG and reverse primer: GGGCACTGTCTTCATCTTAGC). DNA products were separated on 2% agarose gel, and images were taken with NUGenius (Syngene, Frederick, MD, USA).

### 2.8. Mouse Cytokine Array

The Proteome Profiler Mouse XL Cytokine Array (R&D Systems, Minneapolis, MN, USA, Cat. # ARY028) was used to assess cytokines. Frozen pancreatic tissues were suspended in pre-chilled samples in lysis buffer (RIPA buffer, VWR, Radnor, PA, USA, Cat. # 8990, with 100× Halt Protease and Phosphatase inhibitor cocktail (ThermoFisher, Bohemia, NY, USA, Cat. # 78440). The tissues were homogenized with a rotor-stator homogenizer and sonicated three times at 10 s in sonicator QSonica. The extracts were then centrifuged at 10,000× *g* for 10 min at 4 °C and the supernatant was transferred to a new tube. Protein concentration was measured using BCA assay. The cytokine array procedure was performed according to the manufacturer’s instructions. Cytokine arrays were developed and scanned on a c400 imaging system (Azure Biosystems, Dublin, CA, USA, Cat. # c400). Individual dots were inspected for qualitative differences between groups, and quantification was performed using Fiji imaging software (version 2.15.0) [56].

### 2.9. Statistics

The analysis of in vitro and in vivo experiments was performed with *t*-test and One-way ANOVA tests with a value of *p* < 0.05 considered significant. This analysis used GraphPad Prism version 10 for macOS (GraphPad Software, version 10.1.0, www.graphpad.com, accessed on 1 August 2023).

## 3. Results

### 3.1. KLF5 Protein Expression Is Increased in Chronic Pancreatitis

We have previously shown that KLF5 is not present in the normal murine pancreas in pancreatic acinar cells [51]. Additionally, Diaferia et al. showed that KLF5 staining is absent in the normal human pancreas [57]. To confirm these results, we stained normal human and mouse pancreata and showed a lack of KLF5 staining in pancreatic acinar cells and positive staining for KLF5 in human PDAC specimens (Appendix A). To assess the levels of KLF5 in chronic pancreatitis specimens, we stained human tissue microarrays (Figure 1). Here, we showed increased levels of KLF5 within the non-neoplastic epithelial component of pancreatic tissues from patients with chronic pancreatitis (Figure 1).

### 3.2. Inactivation of Klf5 in Pancreatic Acinar Cells’ Reduced Evolution to ADM and PanIN in Chronic Pancreatitis

To investigate whether KLF5 is induced, during the progression of chronic pancreatitis, to pre-neoplastic states, we employed a mouse model with inducible deletion of *Klf5* and/or activation of *KRAS^G12D^* mutant in combination with repetitive pancreatic injury induced by cerulein. H&E-stained tissues of *Ptf1aCre^ERTM^*;*Rosa26^tdTomato/+^* and *Ptf1aCre^ERTM^*;*Rosa26^tdTomato/+^*;*Kras^G12D^* mice treated with cerulein for one, two, three, and four weeks showed significant pancreatic damage compared to control mice (mice treated with PBS for four weeks) (Figure 2). Over the course of four weeks of cerulein treatment, pancreata from *Ptf1aCre^ERTM^*;*Rosa26^tdTomato/+^* presented with characteristics of ADM compared with more extensive ADM and PanIN formation in pancreata from *Ptf1aCre^ERTM^*;*Rosa26^tdTomato/+^*;*Kras^G12D^* mice (Figure 2). Notably, pancreatic tissues of mice with *Klf5* deleted from pancreatic acinar cells show minimal epithelial changes (Figure 2). In addition, pathology assessment confirmed the accumulation of pancreatic damage (edema, necrosis, infiltration of inflammatory cells, loss of pancreatic ducts, ADM, neoplasia) in the *Ptf1aCre^ERTM^*;*Rosa26^tdTomato/+^* and *Ptf1aCre^ERTM^*;*Rosa26^tdTomato/+^*;*Kras^G12D^*. However, the inactivation of *Klf5* significantly decreased pancreatic damage (Figure 3). We demonstrated a significant decrease in ADM and inflammatory infiltration in mice bearing a deletion of *Klf5* compared to appropriate controls (Figure 3B,E). *Ptf1aCre^ERTM^*;*Rosa26^tdTomato/+^*;*Kras^G12D^* mice had increased levels of PanIN compared to other genotypes (Figure 3C). The edema levels were slightly increased in *Ptf1aCre^ERTM^*;*Rosa26^tdTomato/+^* and *Ptf1aCre^ERTM^*;*Rosa26^tdTomato/+^*;*Kras^G12D^* upon chronic injury as compared to mice with *Klf5* deletion (Figure 3D). Additionally, *Ptf1aCre^ERTM^*;*Rosa26^tdTomato/+^* and *Ptf1aCre^ERTM^*;*Rosa26^tdTomato/+^*;*Kras^G12D^* mice had increased loss of pancreatic ducts while *Ptf1aCre^ERTM^*;*Rosa26^tdTomato/+^*;*Klf5^fl/fl^* and *Ptf1aCre^ERTM^*;*Rosa26^tdTomato/+^*;*Kras^G12D^*;*Klf5^fl/fl^* mice did not (Figure 3F). Following four weeks of pancreatic injury with cerulein, we assessed amylase and lipase levels in the serum in *Ptf1aCre^ERTM^*;*Rosa26^tdTomato/+^* and *Ptf1aCre^ERTM^*;*Rosa26^tdTomato/+^*;*Klf5^fl/fl^ mice*. Amylase and lipase levels were increased in the blood of *Ptf1aCre^ERTM^*;*Rosa26^tdTomato/+^* mice treated with cerulein compared to *Ptf1aCre^ERTM^*;*Rosa26^tdTomato/+^*;*Klf5^fl/fl^* mice where an increase was not apparent (Figure 3G,H).

Pancreatic epithelial KLF5 expression during chronic pancreatitis was assessed by immunohistochemistry (IHC). KLF5 expression was increased in the ADM structures in *Ptf1aCre^ERTM^*;*Rosa26^tdTomato/+^* mice, and in the ADM and PanIN structures of *Ptf1aCre^ERTM^*;*Rosa26^tdTomato/+^*;*Kras^G12D^* mice after repetitive injury. KLF5 was not detected in the respective *Klf5* knockout mice and the PBS-treated mice (Figure 4).

### 3.3. Inactivation of Klf5 Reduces ADM and PanIN Formation in Chronic Pancreatitis

To assess the progression of ADM, we performed amylase (pancreatic acinar cell marker) and KRT19 (pancreatic ductal marker) co-staining on pancreata from mice with all four genotypes after PBS and cerulein treatment. Pancreatic tissue obtained from *Ptf1aCre^ERTM^*;*Rosa26^tdTomato/+^* mice treated with cerulein showed co-staining of amylase and KRT19 in injured pancreatic acinar cells confirming metaplastic changes (ADM) (Figure 5 and Appendix A). This increase was also observed in the *Ptf1aCre^ERTM^*;*Rosa26^tdTomato/+^*;*Kras^G12D^* mice upon cerulein treatment, while relatively absent in the respective control mice.

To assess the degree of PanIN formation, we used Alcian blue to highlight mucin-producing cells. We did not observe distinct Alcian blue-positive staining in *Ptf1aCre^ERTM^*;*Rosa26^tdTomato/+^* mice, suggesting that, after four weeks of chronic injury, ADM is the prevalent phenotype in these mice. However, in *Ptf1aCre^ERTM^*;*Rosa26^tdTomato/+^*;*Kras^G12D^* mice, there was a significant increase in Alcian blue-positive cells after chronic injury compared to other conditions (Figure 6 and Appendix A). These results suggest that the deletion of *Klf5* from pancreatic acinar cells prevents ADM and PanIN formation upon chronic injury to the pancreas. In addition, we showed that incomplete deletion of *Klf5* from pancreatic acinar in *Ptf1aCre^ERTM^*;*Rosa26^tdTomato/+^*;*Klf5^fl/fl^* mice results in PanIN formation at a four-week time point (Figure 6, stain marked with an asterisk).

### 3.4. Klf5 Inactivation Reduces Fibrosis in Chronic Pancreatitis

Activation of pancreatic stellate cells is one of the phenomena of chronic pancreatitis which leads to modifications of the pancreatic tissue’s microenvironment and the development and progression of fibrosis [11,12,16,17,19]. To detect the levels of fibrosis, we performed several stains: alpha-SMA [58], Masson’s trichrome [59], and Sirius Red [60]. Pancreatic tissues from both, *Ptf1aCre^ERTM^*;*Rosa26^tdTomato/+^*;*Klf5^fl/fl^* and *Ptf1aCre^ERTM^*;*Rosa26^tdTomato/+^*;*Kras^G12D^*;*Klf5^fl/fl^* mice were negative. Myofibroblasts activation was assessed using an alpha-SMA stain. We observed significant alpha-SMA staining in the pancreatic tissue of the *Ptf1aCre^ERTM^*;*Rosa26^tdTomato/+^* cerulein-treated mice (Figure 7 and Appendix A). The majority of alpha-SMA-positive cells are localized near KRT19-positive ductal cells. There is diffuse staining of alpha-SMA in the pancreas of *Ptf1aCre^ERTM^*;*Rosa26^tdTomato/+^*;*Kras^G12D^*. In contrast, almost no alpha-SMA staining was detected in mice with deleted *Klf5* in pancreatic acinar cells. We quantified the percentage of tissue stained with alpha-SMA (Appendix A), confirming minimal staining in mice with deleted *Klf5* compared with abundant staining in *Ptf1aCre^ERTM^*;*Rosa26^tdTomato/+^* and *Ptf1aCre^ERTM^*;*Rosa26^tdTomato/+^*;*Kras^G12D^* mice.

Masson’s trichrome staining confirmed a high level of fibrosis in *Ptf1aCre^ERTM^*;*Rosa26^tdTomato/+^*;*Kras^G12D^* mice treated with cerulein, focal collagen in *Ptf1aCre^ERTM^*;*Rosa26^tdTomato/+^*-treated mice with cerulein, and no significant staining in pancreatic tissues mice with inactivated *Klf5* in pancreatic epithelial cells in the setting of *Ptf1aCre^ERTM^*;*Rosa26^tdTomato/+^* and *Ptf1aCre^ERTM^*;*Rosa26^tdTomato/+^*;*Kras^G12D^* mice (Figure 8 and Appendix A). These results were confirmed with Picro Sirius Red, which demonstrated scarce collagen I/III in *Ptf1aCre^ERTM^*;*Rosa26^tdTomato/+^* pancreatic tissue obtained from mice treated with cerulein, and a significant production of both collagens in *Ptf1aCre^ERTM^*;*Rosa26^tdTomato/+^*;*Kras^G12D^* mice treated with cerulein using bright field and polarization microscopy (Appendix A). These results confirm that KLF5 is critical in developing injury-related pancreatic fibrosis.

### 3.5. KLF5 Regulates the Expression of Cytokines and Inflammatory Markers in Chronic Pancreatitis

First, to examine whether the inactivation of *Klf5* affects gene expression of inflammatory and fibrotic markers in response to chronic pancreatitis at the RNA level, we collected the whole pancreas from *Ptf1aCre^ERTM^*;*Rosa26^tdTomato/+^* and *Ptf1aCre^ERTM^*;*Rosa26^tdTomato/+^*;*Klf5^fl/fl^* mice after treatment with either PBS or cerulein. The results showed that the levels of *Il6*, *Il1b*, *Tgfb1*, *Tnf*, *Mmp9*, *Timp1*, *Col1a1*, and *Acta2* are increased in *Ptf1aCre^ERTM^*;*Rosa26^tdTomato/+^* mice treated with cerulein as compared to PBS-treated mice (Appendix A). In contrast, these markers were not significantly increased in *Ptf1aCre^ERTM^*;*Rosa26^tdTomato/+^*;*Klf5^fl/fl^* mice after cerulein treatment compared to control mice (PBS-treated) (Appendix A). A direct comparison of the expression levels of the aforementioned markers showed that all but *Tnf* were significantly reduced in *Ptf1aCre^ERTM^*;*Rosa26^tdTomato/+^*;*Klf5^fl/fl^* mice after cerulein treatment as compared to all controls (Figure 9A). To assess the impact of KLF5 on the inflammatory markers in pancreatic acinar cells, we isolated tdTomato-positive cells from *Ptf1aCre^ERTM^*;*Rosa26^tdTomato/+^* and *Ptf1aCre^ERTM^*;*Rosa26^tdTomato/+^*;*Klf5^fl/fl^* mice treated with either PBS or cerulein. The tdTomato-positive cells were sorted, and RNA was isolated as described in the Materials and Methods section. We assessed the levels of *Il6, Il1b*, *Tgfb1*, and *Tnf*. The levels of these markers are increased in pancreatic acinar cells *Ptf1aCre^ERTM^*;*Rosa26^tdTomato/+^* mice treated with cerulein but not in *Ptf1aCre^ERTM^*;*Rosa26^tdTomato/+^*;*Klf5^fl/fl^* cerulein-treated mice as compared to PBS-treated mice (Appendix A). A direct comparison between these mice showed that the deletion of *Klf5* from pancreatic acinar cells significantly reduced their expression upon chronic injury (Figure 9B).

### 3.6. KLF5 Binds to Il1b, Il6, and Tgfb1 Promoters In Vitro and In Vivo

To determine whether KLF5 directly induces *Il1b*, *Il6*, and *Tgfb1* promoters’ activity, we assessed these promoters’ relative activity levels upon *Klf5* overexpression in HEK 293T cells. Our results showed that their activity is significantly induced in HEK 293T cells transfected with *Klf5*-expressing plasmid compared to control cells (Figure 9C). We searched for potential KLF5 binding sites in 1.5 kb sequences upstream of the translation start site of *Il1b*, *Il6*, and *Tgfb1* genes using the JASPAR database [54]. The result showed two adjacent potential binding sites in the *Il1b* and *Il6* promoter regions and three potential binding sites in the *Tgfb1* promoter region (Figure 9D). We then performed a ChIP-PCR assay using HEK 293T cells transfected with *Klf5*-overpressing plasmid and each promoter plasmid. Twenty-four hours after transfection, an anti-KLF5 antibody was used to pull down bound DNA, and PCR was performed using primer sets designed for the potential binding sites. The results showed that KLF5 binds to the promoters of *Il1b*, *Il6*, and *Tgfb1* genes. Due to the proximity of both potential binding sites to the *Il1b* promoter, we could not distinguish between them. Therefore, we used two primers to confirm our results (Figure 9E). Similarly, two KLF5-binding sites in the *Il6* promoter were too close to be distinguished (Figure 9E). In the case of the *Tgfb1* promoter, we could not differentiate between two KLF5-binding sites located in the distal part of the promoter. Furthermore, we performed a ChIP-PCR assay using pancreata from *Ptf1aCre^ERTM^*;*Rosa26^tdTomato/+^* and *Ptf1aCre^ERTM^*;*Rosa26^tdTomato/+^*;*Klf5^fl/fl^* mice treated with cerulein for one week (Figure 10). Our results showed that KLF5 binds to the endogenous promoters of *Il1b*, *Il6*, and *Tgfb1* upon chronic injury. At the same time, in mice with *Klf5* deleted from their pancreatic acinar cells, we could not pull down either of the promoters. In the in vivo model, we pulldown only one tentative KLF5-binding site in the *Tgfb1* promoter. Taken together, we showed that KLF5 could directly bind to *Il1b*, *Il6*, and *Tgfb1* promoters and increase their activity.

### 3.7. Deletion of the Klf5 in Pancreatic Acinar Cells Impacts Inflammatory Response in Chronic Pancreatitis

Chronic pancreatitis is described as a progressive inflammatory disease. Our results showed that deletion of the *Klf5* in pancreatic acinar cells reduces the level of injury to pancreatic acinar cells and the development of fibrosis compared to wild-type mice or mice with activated *KRAS^G12D^* mutation. To assess the impact of KLF5 on the inflammatory response, we performed multiplex mouse cytokine array using pancreata from *Ptf1aCre^ERTM^*;*Rosa26^tdTomato/+^* and *Ptf1aCre^ERTM^*;*Rosa26^tdTomato/+^*;*Klf5^fl/fl^* mice treated with cerulein for four weeks. Our results showed that six proteins are differentially expressed between compared samples. Chemokine (C-C motif) ligands: 6, 11, and 21 (CCL6, CCL11, CCL21, respectively), and Coagulation factor III/Thromboplastin (CD142) levels are reduced, while Adiponectin (ARCP30) and Regenerating Family Member 3 Gamma (RG3G) are slightly increased in *Ptf1aCre^ERTM^*;*Rosa26^tdTomato/+^*;*Klf5^fl/fl^* mice compared to *Ptf1aCre^ERTM^*;*Rosa26^tdTomato/+^* mice upon cerulein treatment. (Figure 11A,B). In summary, deletion of the *Klf5* in pancreatic acinar cells reduces inflammatory response upon chronic injury to the pancreas.

## 4. Discussion

In this study, we showed that KLF5 is essential during the development and progression of chronic pancreatitis. We demonstrated that KLF5 levels are increased in the pancreas, originating from patients with chronic pancreatitis and from a murine model of chronic pancreatitis (Figure 1 and Figure 4). Notably, we showed that deletion of the *Klf5* in pancreatic acinar cells by itself or in the context of *KRAS^G12D^* mutant inhibits chronic pancreatitis development (Figure 2 and Figure 3 and Appendix A). In our studies, we used two genetic models, *Ptf1aCre^ERTM^*;*Rosa26^tdTomato/+^* and *Ptf1aCre^ERTM^*;*Rosa26^tdTomato/+^*;*Kras^G12D^* mice, in combination with repetitive injury caused by intraperitoneal injections of cerulein. Owing to the *Ptf1a* promoter, these mice expressed the CreERTM recombinase in pancreatic acinar cells. Upon tamoxifen treatment, *Kras^G12D^* expression is activated through CreERTM-loxP recombination only in the adult pancreatic acinar cells. Following an induction regimen of intraperitoneal injections of tamoxifen or vehicle (corn oil) every other day for 6 days, the *Ptf1aCre^ERTM^*;*Rosa26^tdTomato/+^*;*Kras^G12D^* and *Ptf1aCre^ERTM^*;*Rosa26^tdTomato/+^* (control) mice were subjected to chronic pancreatic insult by repeated 6-hourly intraperitoneal injections of cerulein or PBS, three days a week, for four consecutive weeks. This regimen led to the development of mild chronic pancreatitis with pancreatic histology in *Ptf1aCre^ERTM^*;*Rosa26^tdTomato/+^* control mice with preserved pancreatic acinar cells, ADM, and scarce fibrosis. Notably, in *Ptf1aCre^ERTM^*;*Rosa26^tdTomato/+^*;*Kras^G12D^* mice, the same injury regimen led to the development of advanced chronic pancreatitis with early neoplasia, as characterized by extensive fibrosis and insufficiency of pancreatic exocrine functions [5,61]. In our studies, the pancreatic tissues of *Ptf1aCre^ERTM^*;*Rosa26^tdTomato/+^*;*Kras^G12D^* mice treated with cerulein are almost entirely fibrotic and devoid of pancreatic acinar cells, and thus, the levels of amylase and lipase are almost non-existent. Therefore, we only assessed the levels of these enzymes in *Ptf1aCre^ERTM^*;*Rosa26^tdTomato/+^* and *Ptf1aCre^ERTM^*;*Rosa26^tdTomato/+^*;*Klf5^fl/fl^* mice treated with PBS and cerulein. Our results showed that the amylase and lipase levels in *Ptf1aCre^ERTM^*;*Rosa26^tdTomato/+^* mice treated with cerulein are slightly increased compared to PBS-treated mice. In addition, the inactivation of *Klf5* significantly reduced serum amylase and lipase levels in our model (Figure 3G,H). This could be because we collected samples for enzyme analysis three days after the final injection of cerulein or PBS and, thus, allowed the tissue to resolve from the acute injury. In most cases of human pancreatitis and murine models of chronic pancreatitis with extensive fibrosis, the levels of these enzymes are not changed or reduced [62,63]. However, under mild conditions, it has been previously shown that levels of both enzymes can be increased in the murine model of chronic pancreatitis and then decreased as the disease progresses [64,65,66,67].

Our previous study showed that KLF5 is crucial for ADM and PanIN formation during acute injury and spontaneous PanIN formation upon singular *KRAS^G12D^* activation [51]. Here, we demonstrate that KLF5 is necessary for the development of ADM and PanIN lesions during recurrent pancreatic injury. Deletion of the *Klf5* in pancreatic acinar cells by itself or in the context of KRAS^G12D^ mutant and in combination with recurrent injury reduces the development of ADM and PanIN lesions (Figure 5 and Figure 6 and Appendix A). KLF transcription factors have been previously shown to play an essential role in acute pancreatitis development and progression. Similarly to the inhibition of *Klf5,* the inactivation of *Klf4* in the setting of the acute injury to the pancreas and the presence of *KRAS^G12D^* mutation led to reduced levels of ADM and PanIN [68]. Conversely, *Klf4* overexpression induced ADM and caused an increase in the ductal markers’ expression [68]. Furthermore, KLF4 and KLF5 have been shown to play a crucial role in pancreatic carcinogenesis and regulate processes such as proliferation and epithelial-to-mesenchymal transition [51,69,70,71]. These studies suggest that KLF4 can regulate ADM and PanIN formation processes during chronic pancreatitis, utilizing similar mechanisms as in acute injury.

The microenvironment of chronic pancreatitis tissues is characterized by massive fibrosis. In healthy pancreatic tissues, pancreatic stellate cells are in quiescent status. However, during chronic injury, injured pancreatic acinar cells and immune cells produce multiple factors that cause the activation of pancreatic stellate cells and lead to fibrosis [10,16,18,19]. Masson’s Trichome, Sirius Red, and alpha-SMA stains in *Ptf1aCre^ERTM^*;*Rosa26^tdTomato/+^* showed a slight development of fibrosis, and, in *Ptf1aCre^ERTM^*;*Rosa26^tdTomato/+^*;*Kras^G12D^*, a major remodeling of the pancreatic tissue that is characterized by fibrosis [72]. Our data show that the inactivation of *Klf5* significantly repressed stellate cell activation and fibrosis formation in the control mice and the context of activated *KRAS^G12D^* (Figure 7 and Figure 8 and Appendix A). Furthermore, our in vitro and in vivo studies indicate that KLF5 regulates the expression levels of cytokines and fibrotic components such as *Il6*, *Il1b*, *Tgfb1*, *Tnf, Col1a1*, *Acta2*, *Timp1*, and *Mmp9* (Figure 9 and Appendix A) when analyzing whole pancreatic tissues from in *Ptf1aCre^ERTM^*;*Rosa26^tdTomato/+^* and *Ptf1aCre^ERTM^*;*Rosa26^tdTomato/+^*;*Klf5^fl/fl^* mice treated with cerulein as compared to appropriate controls. The decreased levels of fibrosis are probably secondary effects of the deletion of *Klf5* from pancreatic acinar cells. Indeed, isolation of pancreatic acinar cells (tdTomato-positive cells) showed decreased levels of markers (*Il6*, *Il1b*, *Tgfb1*, *Tnf*) commonly expressed by injured pancreatic acinar cells [17,73,74,75,76,77,78,79,80]. As multiple publications show that KLF5 regulates *Tnf* expression, we focused on investigating the activation of *Il6*, *Il1b*, and *Tgfb1* by KLF5 [81,82]. Our results showed that overexpression of *Klf5* leads to increased activity of the promoters of the genes above. We have provided evidence that KLF5 directly binds to the specific sites in their promoters in vitro and in vivo and drives their expression (Figure 9 and Figure 10). Chronic pancreatitis is characterized by high levels of inflammation and progressive fibrosis and is often called fibroinflammatory disease [83,84]. Here, we showed that the deletion of *Klf5* results in decreased levels of several vital chemokines, such as CCL6, CCL11, and CCL21 (Figure 11). Studies showed that CCL6 plays an essential role in the pathogenesis of pulmonary fibrosis, and its levels have been previously shown to increase upon chronic pancreatitis. CCL6 is a chemoattractant for macrophages, monocytes, and T cells [85,86,87]. CCL11 (eotaxin) was demonstrated to attract multiple immune cells such as eosinophils, neutrophils, and Th2 lymphocytes; however, in the context of IL5, it has been previously shown that eosinophils increase upon chronic injury to the pancreas [88,89]. Notably, CCL11 activation was demonstrated as a result of Toll-Like Receptor 9 (TLR9) activation in PDAC development and progression [90]. CCL21 induces CD133^+^ pancreatic cancer, stem-like cell metastasis, and the migration of pancreatic tumor cells [91,92]. Interestingly, the cytokine analysis identified CD142, a pancreatic embryonic progenitor marker, as being decreased upon *Klf5* deletion during chronic injury (Figure 11). CD142 increased during dedifferentiation of pancreatic acinar cells that, afterward, transdifferentiated to duct-like cells [93]. Studies by Araki and colleagues have shown that exogenous adiponectin was protective in acute pancreatitis in mice fed with a high-fat diet [94]. In multiple studies, adiponectin is anti-inflammatory, and its deficiency increases the inflammatory response in various mouse models [95,96]. On the other hand, mice with deleted *Klf5* showed increased levels of REG3G. REG3G provides defense against bacterial activity in the gut. It is protective against colitis, diabetic wound healing, and alcohol-induced fatty liver disease by suppressing epithelial inflammation [97]. However, its role in pancreatitis has yet to be well studied. One study showed that exogenous delivery of REG3G contributes to the progression of chronic pancreatitis towards neoplasia [98]. At the same time, another study showed that it ameliorates pancreatic β-cell dysfunction in mice [99]. Previous publications demonstrated that Cre recombinase activity is increased in female mice compared to male mice [37,100] and that tamoxifen treatment may have a different effect on female and male mice [101,102,103,104]. Despite the differences in response to the tamoxifen between female and male mice, we demonstrated similar results in both experimental groups and, thus, presented them together.

## 5. Conclusions

In summary, our results support the role of KLF5 in developing ADM and PanIN lesions in chronic pancreatic injury and provide evidence that the deletion of *Klf5* from pancreatic acinar cells reduces remodeling in the pancreatic microenvironment. Moreover, we showed that KLF5 directly regulates and binds to the promoters of *Il1b, Il6*, and *Tgfb1*, and simultaneously, its deletion results in decreased levels of multiple inflammatory and fibrotic markers in *Ptf1aCre^ERTM^*;*Rosa26^tdTomato/+^*;*Klf5^fl/fl^* and *Ptf1aCre^ERTM^*;*Rosa26^tdTomato/+^*;*Kras^G12D^*;*Klf5^fl/fl^* mice after repetitive injury. An in-depth understanding of the role of KLF5 during the progression of pancreatic injury may become attractive for the development of therapeutics for chronic pancreatitis treatment.

## Figures and Tables

**Figure 1 cancers-15-05427-f001:**
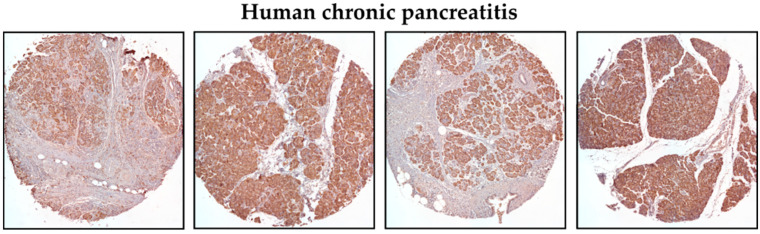
Representative immunohistochemistry images of KLF5 expression in the pancreas of chronic pancreatitis patients (magnification 10×).

**Figure 2 cancers-15-05427-f002:**
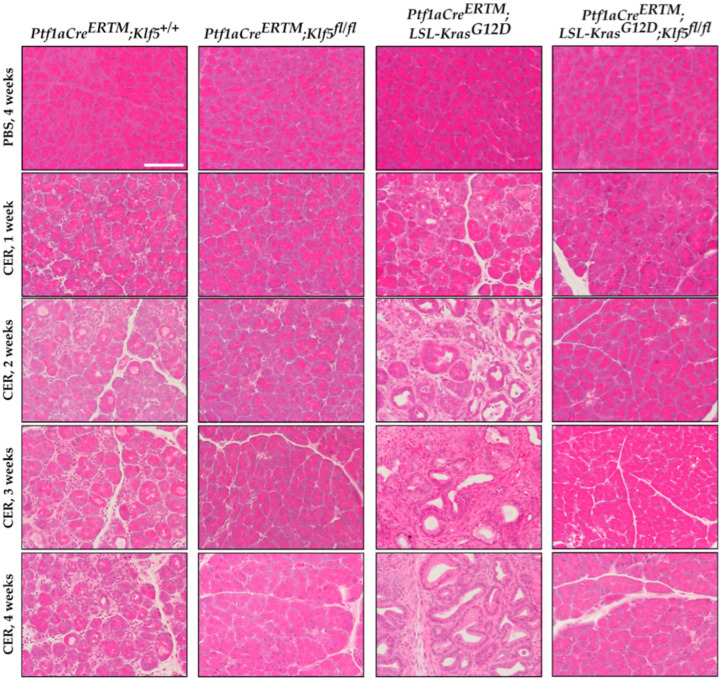
Deletion of *Klf5* in pancreatic acinar cells results in the inhibition of chronic pancreatitis in a murine model of this disease. Representative images of H&E staining of the pancreas of normal and chronic pancreatitis mice. *Ptf1aCre^ERTM^*;*Rosa26^tdTomato/+^*, *Ptf1aCre^ERTM^*;*Rosa26^tdTomato/+^*;*Klf5^fl/fl^*, *Ptf1aCre^ERTM^*;*Rosa26^tdTomato/+^*;*Kras^G12D^*, and *Ptf1aCre^ERTM^*;*Rosa26^tdTomato/+^*;*Kras^G12D^*;*Klf5^fl/fl^* mice were treated with cerulein for one, two, three, and four weeks or with PBS for four weeks. CER—cerulein. Scale bar = 200 µm.

**Figure 3 cancers-15-05427-f003:**
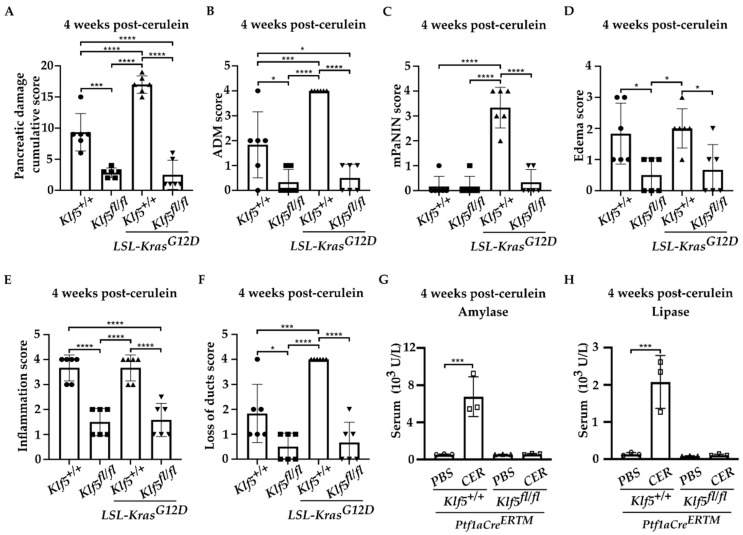
Deletion of *Klf5* in pancreatic acinar cells reduces injury during chronic pancreatitis in a murine model of this disease. *Ptf1aCre^ERTM^*;*Rosa26^tdTomato/+^*, *Ptf1aCre^ERTM^*;*Rosa26^tdTomato/+^*;*Klf5^fl/fl^*, *Ptf1aCre^ERTM^*;*Rosa26^tdTomato/+^*;*Kras^G12D^*, and *Ptf1aCre^ERTM^*;*Rosa26^tdTomato/+^*;*Kras^G12D^*;*Klf5^fl/fl^* mice were treated with cerulein or PBS for four weeks. CER—cerulein. (**A**) Quantification of pancreatic damage score per pancreas after cerulein treatment (*n* = 6). (**B**) ADM scores, (**C**) mPanIN score, (**D**) edema, (**E**) inflammatory infiltration, (**F**) loss of pancreatic ducts per pancreas from female mice after cerulein treatment. * *p* < 0.05, *** *p* < 0.001, and **** *p* < 0.0001 using one-way ANOVA test (Data represent mean ± S.D.). Quantification of serum amylase (**G**) and lipase (**H**) from cerulein-treated and PBS-treated mice (*n* = 3). *** *p* < 0.001 by *t*-test (Data represent mean ± S.D.).

**Figure 4 cancers-15-05427-f004:**
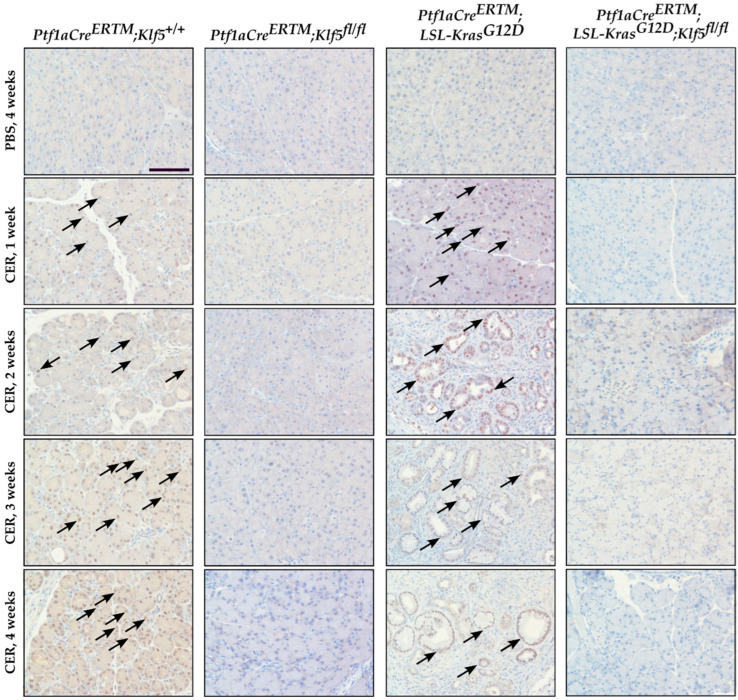
KLF5 levels are increased in the murine model of chronic pancreatitis. Immunohistochemical analysis of KLF5 expression in pancreatic tissues from female mice of indicated genotypes and treatments. Scale bars = 200 µm. Arrows show examples of KLF5-positive staining.

**Figure 5 cancers-15-05427-f005:**
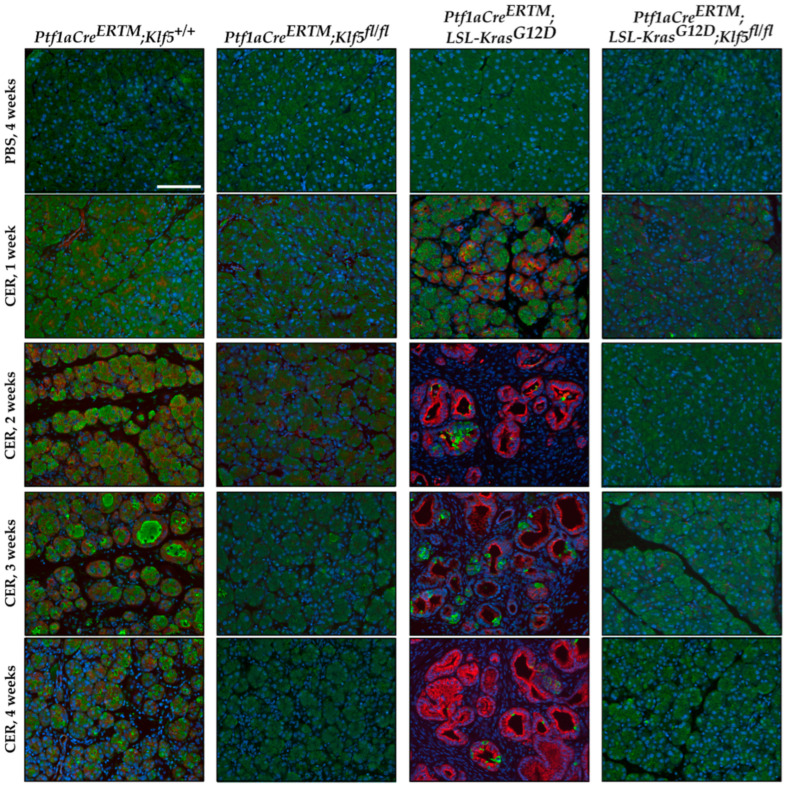
Genetic inactivation of *Klf5* inhibits the transformation of acinar cells to ADM following chronic injury. Multicolor immunofluorescence (IF) analysis of amylase (green), Keratin-19 (red), and nuclei (Blue) in pancreatic tissues from mice of indicated genotypes and treatments. Scale bar = 200 µm.

**Figure 6 cancers-15-05427-f006:**
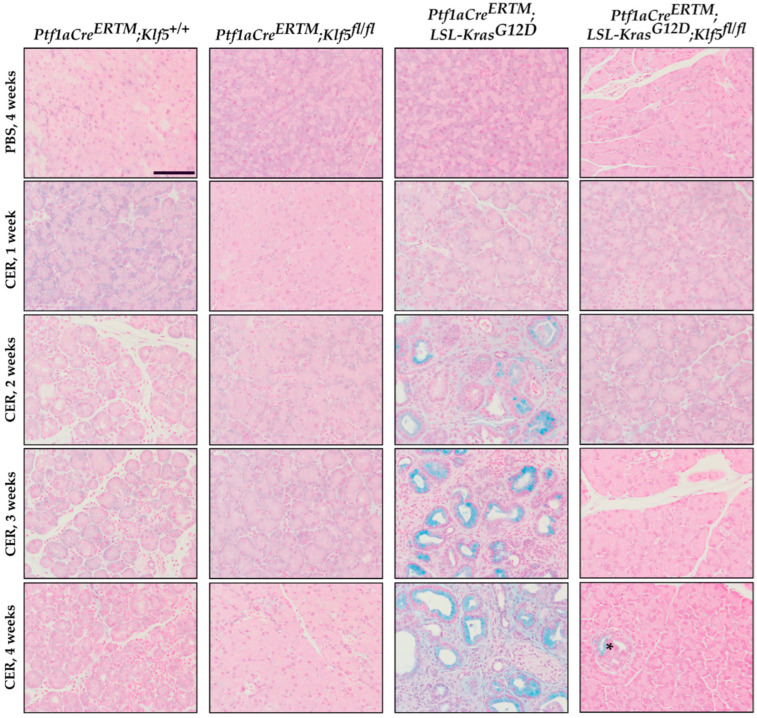
Genetic inactivation of *Klf5* inhibits PanIN formation following chronic injury. Alcian blue staining counterstained with Nuclear Fast Red of pancreatic tissue from mice of indicated genotypes and treatments. Scale bar = 200 µm. * marks positive Alcian blue stain in *Ptf1aCre^ERTM^*;*Rosa26^tdTomato/+^*;*Klf5^fl/fl^* mice.

**Figure 7 cancers-15-05427-f007:**
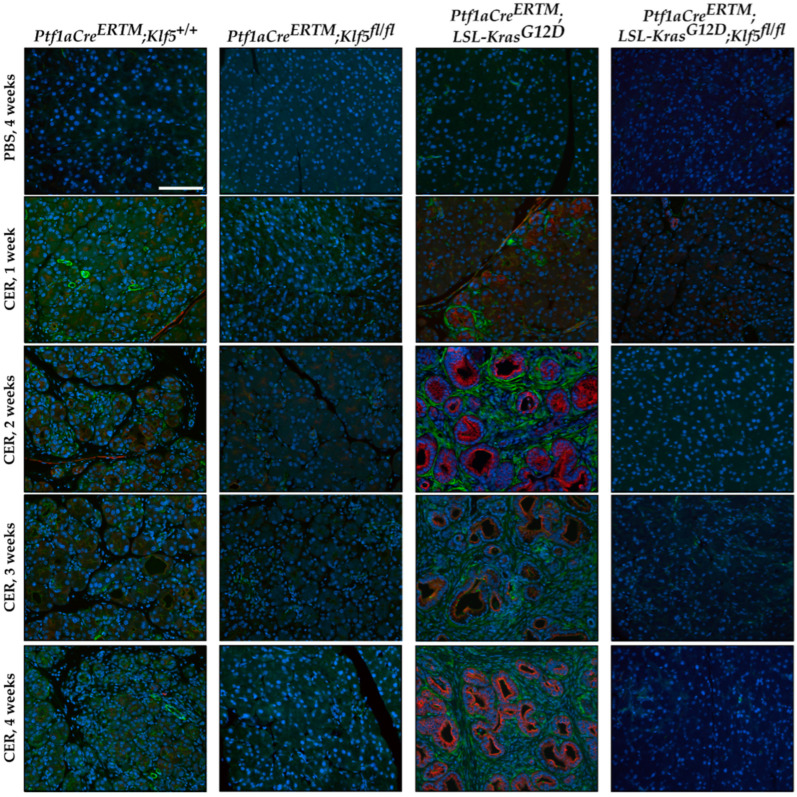
Genetic inactivation of *Klf5* in vivo suppresses pancreatic stellate cell activation. Representative images of multicolor immunofluorescence (IF) analysis of alpha-SMA (green), Keratin-19 (red), and nuclei (Blue) in pancreatic tissues from mice of indicated genotypes and treatments. Scale bar = 200 µm.

**Figure 8 cancers-15-05427-f008:**
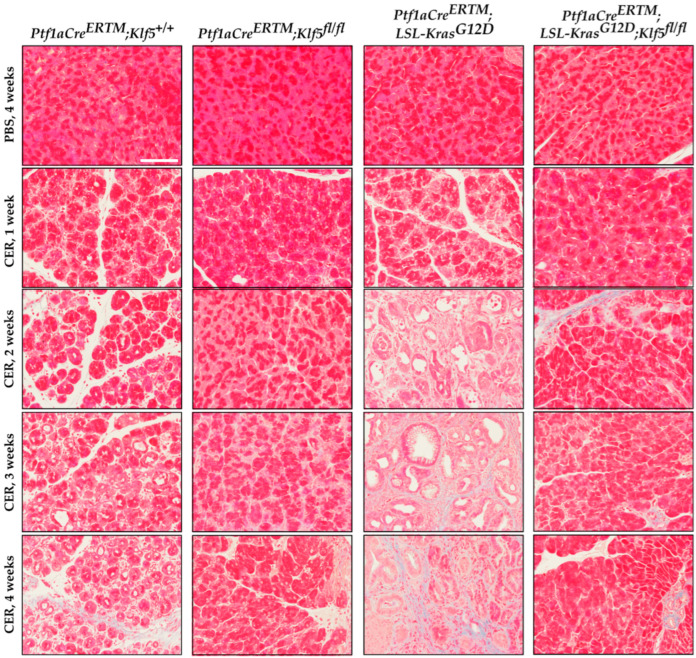
Genetic inactivation of *Klf5* in vivo suppresses fibrosis. Representative images of Masson’s trichrome stain of mice of indicated genotypes and treatment. Scale bar = 200 µm.

**Figure 9 cancers-15-05427-f009:**
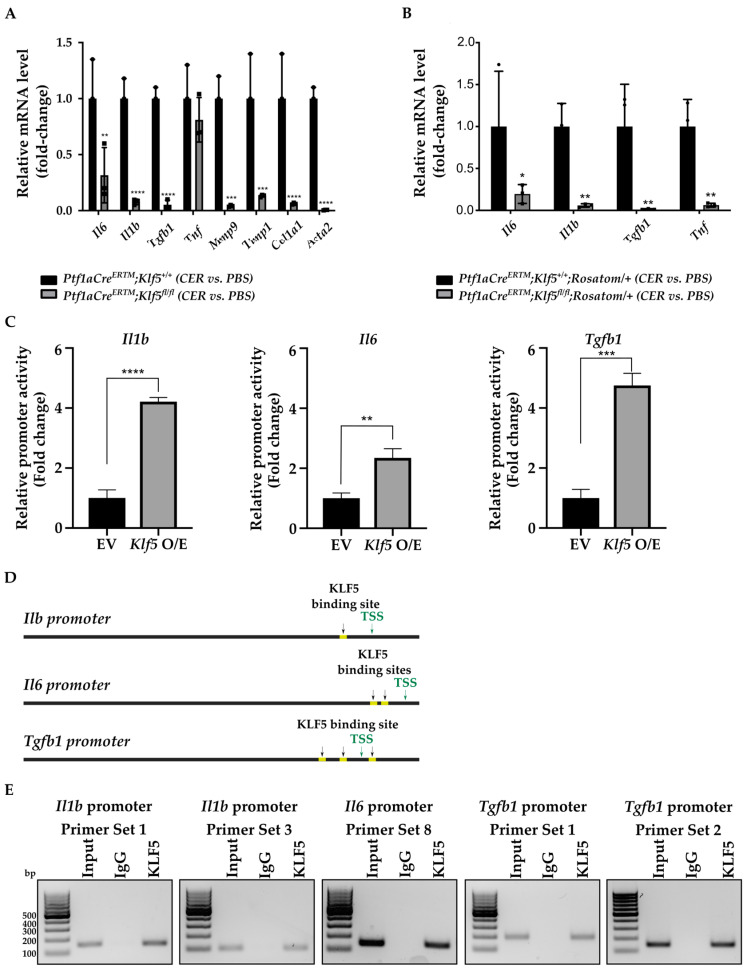
KLF5 regulates the expression of inflammatory and fibrotic markers during chronic pancreatitis. (**A**) qRT-PCR results of gene expression of *Il6*, *Il1b*, *Tgfb1*, *Tnf*, *Mmp9*, *Timp1*, *Col1a1*, and *Acta2* in the whole pancreatic tissues from mice of indicated genotypes and treatments. ** *p* < 0.01, *** *p* < 0.001, **** *p* < 0.0001, from one-way ANOVA test (Data represent mean ± S.D.). (**B**) qRT-PCR results of gene expression of *Il6*, *Il1b*, *Tgfb1*, and *Tnf* in tdTomato-positive cells originated from pancreatic tissues from mice of indicated genotypes and treatments. * *p* < 0.05, ** *p* < 0.01 from one-way ANOVA test (*n* = 3, data represent mean ± S.D.). (**C**) Relative activity of *Il1b*, *Il6*, and *Tgfb1* promoters. HEK 293T cells were co-transfected with control plasmid or plasmid encoding KLF5 and either of the promoters. Twenty-four hours after transfection, the activity of the promoters was measured. ** *p* < 0.01, *** *p* < 0.001, **** *p* < 0.0001 by *t*-test (*n* = 6, Data represent mean ± SD). (**D**) Predicted binding sites (yellow) for KLF5 in sequence 1.5 kb upstream of the translation start site of the *Il1b*, *Il6*, and *Tgfb1* promoters using the JASPAR database. T.S.S.—transcription start site. (**E**) PCR amplification of site-specific sequences of DNA. Product of ChIP from *Il1b*, *Il6*, and *Tgfb1* promoters using an anti-KLF5 antibody.

**Figure 10 cancers-15-05427-f010:**
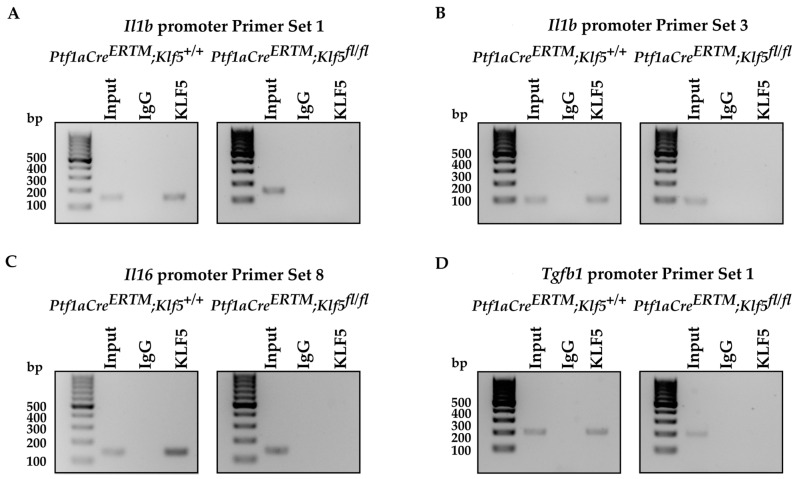
KLF5 binds to the promoters of *Il6*, *Il1b*, and *Tgfb1* during chronic pancreatitis. PCR amplification of site-specific sequences of DNA. Product of ChIP-PCR from (**A**–**D**) *Il1b*, *Il6*, and *Tgfb1* promoters using an anti-KLF5 antibody.

**Figure 11 cancers-15-05427-f011:**
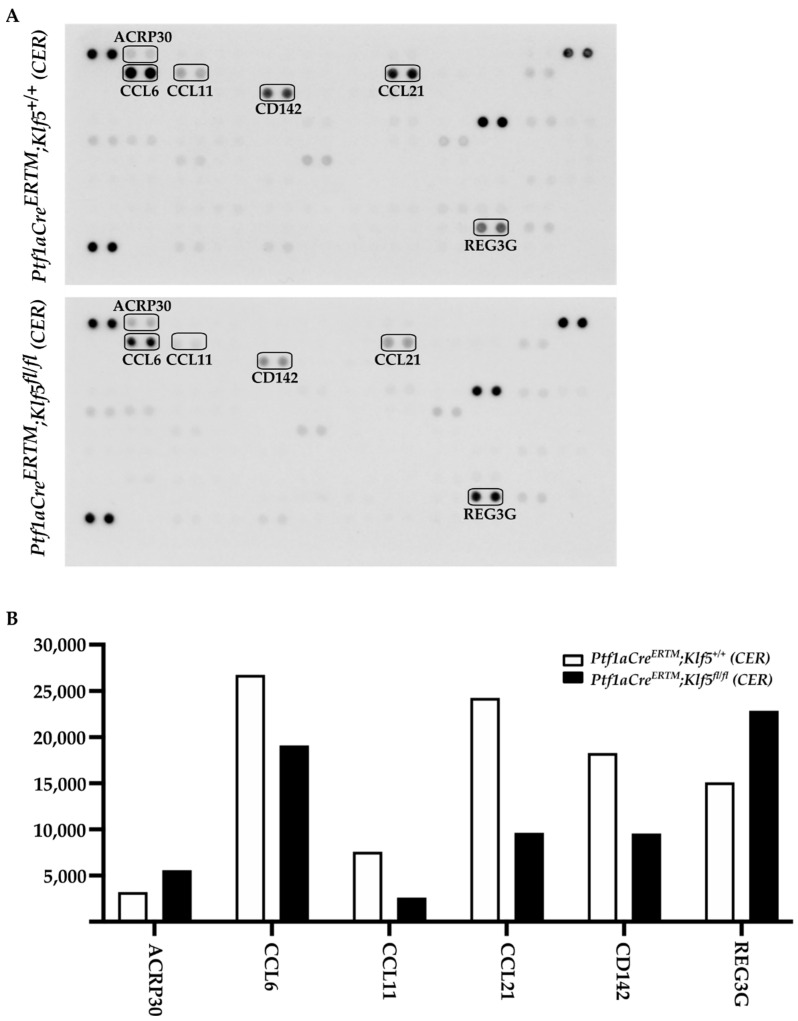
Deletion of *Klf5* in vivo inflammatory response upon chronic pancreatitis. (**A**) Mouse cytokine arrays showing the levels of proteins from pancreatic tissues of *Ptf1aCre^ERTM^*; *Rosa26^tdTomato/+^* and *Ptf1aCre^ERTM^*; *Rosa26^tdTomato/+^*; *Klf5^fl/fl^* mice treated with cerulein for four weeks. The boxes mark differentially expressed proteins. (**B**) Quantification of the levels of differentially expressed proteins from (**A**).

## Data Availability

The data presented in this study are available in this article (and Appendix A).

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
