# Peer review of "Krüppel-like Factor 5 Plays an Important Role in the Pathogenesis of Chronic Pancreatitis"

_cancers, 2023, doi:10.3390/cancers15225427_

Round 1
Reviewer 1 Report (Previous Reviewer 1)
Comments and Suggestions for Authors
We are satisfied with the responses provided by the authors. I have no further issues with the manuscript other than a few misspellings, which should be picked up by the copy-editing process.
Comments on the Quality of English LanguageI have no further issues with the manuscript other than a few misspellings, which should be picked up by the copy-editing process.
Author Response
We want to thank you, the Reviewer, for your comments. We will spell-check the manuscript and submit an updated version.
Reviewer 2 Report (New Reviewer)
Comments and Suggestions for Authors
The manuscript Krüppel-Like Factor 5 Plays an Important Role in the Pathogenesis of Chronic Pancreatitis by Alavi et al is talking about the Krüppel-Like Factor 5 in Pancreatitis. The authors have used in vivo and in vitro models to prove the concept. The manuscript is well written and presented by the figures. The point-wise comments are as follows.
1. For the in vitro work, the authors have used the HEK 293T, kidney cell line. Why a kidney cell lines used in a Pancreatitis study?
2. In Figure 7 authors tested aSMA for fibrosis induction. Did, other fibrotic markers were checked for fibrosis induction?
3. What is the rationale for choosing KLF 5 in this study, is not clear.
Author Response
We want to thank the Reviewers for their comments. Please see our response below.
Comment 1. For the in vitro work, the authors have used the HEK 293T kidney cell line. Why are kidney cell lines used in a Pancreatitis study?
Response 1. Initially, we performed ChIP-PCR in HEK cells, and we were asked by one of the reviewers to add data from in vivo studies. Thus, the current manuscript contains original results and data from the in vivo model. We were not asked to remove the initial set of the results.
Comment 2. In Figure 7, the authors tested aSMA for fibrosis induction. Did other fibrotic markers were checked for fibrosis induction?
Response 2. We performed staining for which collagen I/III. The results are included in Supplementary Figure 2.
Comment 3. What is the rationale for choosing KLF5 in this study is not clear.
Response 3. Others and we showed that KLF5 plays a vital role in pancreatic cancer development and maintenance (PMID: 19671674, 24041470, 26769127, 28864900, 29248441, 32985124, 35104421). In addition, we showed that KLF5 is critical during the response to acute pancreatitis (PMID: 29248441). The current study expands upon previous results and demonstrates KLF5's role in chronic pancreatitis.
Reviewer 3 Report (New Reviewer)
Comments and Suggestions for Authors
Results of this study are significant and relevant for understanding pathogenesis of chronic pancreatitis in association with Klf5.
Author Response
We want to thank you, the Reviewer, for your comments.
Reviewer 4 Report (New Reviewer)
Comments and Suggestions for Authors
Chronic pancreatitis has been recognized as an important risk for pancreatic cancer. However, its mechanism remains unclear. This article is based on and continues research previously reported by the authors (Ref. 51). Their previous studies showed that KLF5 is critical for the formation of ADM and PanIN during acute injury and promotes proliferation, acinar-to-ductal metaplasia, and tumor growth in mice. In this study, enhanced expression of KLF5 was demonstrated in pancreatic sections from pancreatitis patients and cerulein-treated mice. Using conditional knockout mice, the authors also demonstrated that the presence of KLF5 is associated with pancreatitis and contributes to ADM and PanIN lesions. Deletion of Klf5 also significantly reduced a group of inflammation-related genes, including Il1b, Il6, Tgfb1, Timp1, and Mmp9. Furthermore, the final results showed the binding activity of KLF5 to the promoters of most of these genes. This study provides important information on the pathological markers of pancreatitis and its pathological mechanisms. The results here are significant and detailed, showing how pancreatitis progresses to precancerous pancreatic cancer through KLF5 expression.
The following is my comment.
As we all know, cerulein is a ten-amino acid peptide extracted from Australian green frogs. It is similar to gastrin and CCK and binds to CCK-R2. Although cerulein injection is one of the most representative and widely used experimental models, the pathway from the clinical etiology of pancreatitis to KLF5 expression, as well as the role of gastrin or CCK in pancreatitis, remains unclear. Whether other risk factors, including alcohol abuse, hypertriglyceridemia, smoking, and microbiota dysbiosis, upregulate KLF5 expression also needs to be clarified in the future.
Author Response
We want to thank you, the Reviewer, for your comments. We agree with the Reviewer's comments that future studies employing different etiologies of chronic pancreatitis will be helpful to understand KLF5's role in pancreatitis comprehensively.
This manuscript is a resubmission of an earlier submission. The following is a list of the peer review reports and author responses from that submission.
Round 1
Reviewer 1 Report
Comments and Suggestions for Authors
Transcription factor KLF5 has been studied by the present investigators and some remarkable findings have been reported previously on its role in the GI tract. There is a persistent misconception of its expression, e.g., “only in the testis and placenta” (PhosphoSitePlus), although several studies have evaluated its influence on distinct cancers. The current manuscript follows up previous work analyzing the role of KLF5 in PDAC associated disease progression and associated pathologic transformations such as pancreatic stellate cell activation, fibrosis, ADM and PanIN formation in the pancreas of genetically engineered mice of the KC genotype. The present paper is well written and addresses the role of KLF5 in chronic pancreatitis (CP) development with and without the mutant Kras, that was not previously addressed. KrasG12D expression floxed KLF5 and additional genotypes including Tomato expression are induced using a tamoxifen-inducible variant of Ptf1-Cre. The work is of high quality. However, some concerns, and a few minor omissions and typos that were detected should be addressed in a revised version.
Major questions
1. By and large the experiments are clearly and succinctly described. However, it is not completely clear what age the animals are at different points during the CP experiment. If their ages are slightly different, this should be mentioned, and a range given. This is important for the field to understand, because the transformations being described take time and this should be carefully documented. So, what age are the mice when tamoxifen is administered. How long after tamoxifen administration does the CP get started. The length of treatment itself is well defined and mice are sacrificed “three days after…” so that part is clear.
2. The tamoxifen treatment performed is reported without mentioning any controls that might have been performed now or before that would consider the possibility of tamoxifen-dependent events. There are no tamoxifen controls and why they were not thought to be necessary is not specified. If tamoxifen is quickly metabolized, this could be mentioned. This point is important, as KLF5 may cooperate with estrogen receptors, as was reported in e.g., prostate and breast cancers, and which are potential targets of tamoxifen. This reviewer consulted an expert on the treatment and was told the tamoxifen concentrations used are relatively high, and corn oil residue could be present in the mice peritoneum even after extended periods. Tamoxifen can be administered by gavage, please briefly provide any available details on whether this method was attempted.
3. The authors maintain that examination of KLF5 staining in the CP patients in Fig. 1 indicates increased KLF5, however no controls (staining from patients without CP are shown, albeit such tissues may not be readily available). In the Human Atlas website, potential for detecting nuclear KLF5 in the normal pancreas tissue is evident, possibly increasing at more advanced age. Please clarify the basis for the premise that normal tissue does not express the factor, or expresses less than in cases of CP. Simply stating that it was shown in the previous paper (I could not find it there) is not sufficient. There is no need to assert that KLF5 is never expressed in human acinar cells, so such language should be moderated.
4. There seem to be some redundant data. Either extra examples of the same data (e.g., 1B, supplementary 1A, etc.) can be further consolidated into summary data or they can be simply removed and/or described as further data not shown.
5. The presentation of amylase/lipase data is somewhat confusing. In cerulein-based rodent models of acute pancreatitis, theory suggests that these digestive enzymes are detected in the blood due to supraphysiologic hyperstimulation of the acinar cells that leads to aberrant basolateral secretion. Chronic pancreatitis has been modeled previously in rodents by three times weekly bouts of multiple cerulein for six weeks, with significant fibrosis being detected. In the present work, whereas some fibrotic changes are evident after only four weeks, the disease must be considered early, rather than late chronic CP as during late CP, due to accumulated acinar cell damage and fibrosis, loss of digestive function can be a factor and thus, the presence of robust serum levels of digestive enzymes following cerulein administration could no longer be assumed. As such, it is not clear which aspect of KLF5 deletion is responsible for the failure to obtain elevated blood levels of amylase and lipase in the KLF5-floxed mice with much less accumulated damage. The authors should try to reconcile this rhetorically, and/or substantiate KLF5-dependent reasons for the acinar cells to be unresponsive to the cerulein administration.
Further minor comments.
L265, in the blood of mice, not “in the mice” should be emphasized.
L450-1, “extensive” fibrosis in KRasG12D mice seems exaggerated to say at this early time point.
L457, The sentence seems incomplete. “..treated”
l.459, please use either “deletion of Klf5” OR “Klf5 inactivation” as using both is redundant
l459, suggest, “Indeed” rather than “However”
l465, prefer “provide evidence that” to “demonstrate”
l481, promoters
Author Response
October 14th, 2022
Re: Cancer-1956515
We want to thank the Reviewer for his/her comments and suggestions. Please find our set-by-step response included below.
Major questions.
Comment 1. By and large the experiments are clearly and succinctly described. However, it is not completely clear what age the animals are at different points during the CP experiment. If their ages are slightly different, this should be mentioned, and a range given. This is important for the field to understand because the transformations being described take time and this should be carefully documented. So, what age are the mice when tamoxifen is administered. How long after tamoxifen administration does the CP get started. The length of treatment itself is well defined and mice are sacrificed “three days after…” so that part is clear.
Response 1. We thank the Reviewer for this comment. The experiments started (tamoxifen injections) when the mice were 6-8 weeks old. The tamoxifen injections were done on Monday, Wednesday, and Friday), and on the following Monday, we started CP treatment. Therefore, we modified our statement in the Material and Method section (2.1 Animal studies).
“We employed 8- to 12-week-old and gender-matched mice in this study. To induce Cre-mediated recombination, we performed three intraperitoneal injections every other day of tamoxifen (Sigma-Aldrich, T5648; 3mg/injection) dissolved in corn oil on week 1. Corn oil-injected mice were used as controls. Chronic pancreatitis was induced one week after the first tamoxifen injection by I.P. injections of 100µl cerulein (BACHEM, H3220) at 50 µg/kg of body weight in DPBS (Fisher Scientific, 21-031-CV) six times a day, three times a week for four weeks.”
Comment 2. The tamoxifen treatment performed is reported without mentioning any controls that might have been performed now or before that would consider the possibility of tamoxifen-dependent events. There are no tamoxifen controls and why they were not thought to be necessary is not specified. If tamoxifen is quickly metabolized, this could be mentioned. This point is important, as KLF5 may cooperate with estrogen receptors, as was reported in e.g., prostate and breast cancers, and which are potential targets of tamoxifen. This Reviewer consulted an expert on the treatment and was told the tamoxifen concentrations used are relatively high, and corn oil residue could be present in the mice peritoneum even after extended periods. Tamoxifen can be administered by gavage, please briefly provide any available details on whether this method was attempted.
Response 2. We thank the Reviewer for this comment. We utilized the concentration of tamoxifen that has been previously published in the same animal model, "Spatiotemporal patterns of multipotentiality in Ptf1a-expressing cells during pancreas organogenesis and injury-induced facultative restoration” Fong Cheng Pan, Eric D. Bankaitis, Daniel Boyer, Xiaobo Xu, Mark Van de Casteele, Mark A. Magnuson, Harry Heimberg, Christopher V. E. Wright. Development (2013) 140 (4): 751–764. https://doi.org/10.1242/dev.090159. The same concentration of tamoxifen has been used by our laboratory and other laboratories and published (e.g., PMID: 29248441, 28087712, 32709695, 31422917). This method allows for achieving around 60-80% recombination efficiency in studied mice (male mice being on the lower end and female mice being on the higher end of recombination). We tested a lower concentration of tamoxifen but did not achieve efficient recombination using our system. The half-life of tamoxifen has been established to be between 6.8 hours and -11 hours (PMID: 25318936). We used 100ul of tamoxifen in corn oil in IP injections. This is the standard volume used for published studies.
Comment 3. The authors maintain that examination of KLF5 staining in the CP patients in Fig. 1 indicates increased KLF5, however no controls (staining from patients without CP are shown, albeit such tissues may not be readily available). In the Human Atlas website, potential for detecting nuclear KLF5 in the normal pancreas tissue is evident, possibly increasing at more advanced age. Please clarify the basis for the premise that normal tissue does not express the factor, or expresses less than in cases of CP. Simply stating that it was shown in the previous paper (I could not find it there) is not sufficient. There is no need to assert that KLF5 is never expressed in human acinar cells, so such language should be moderated.
Response 3. We want to thank the Reviewer for this comment. Unfortunately, we were not able to obtain normal human pancreatic tissues. Therefore, the citation provided refers to a normal pancreas obtained from mice.
The correct publication is entitled “Dissection of transcriptional and cis-regulatory control of differentiation in human pancreatic cancer." Giuseppe R Diaferia, Chiara Balestrieri, Elena Prosperini, Paola Nicoli, Paola Spaggiari, Alessandro Zerbi, Gioacchino Natoli. The EMBO Journal (2016)35:595-617https://doi.org/10.15252/embj.201592404. The authors of this publication showed in Figure 5A that KLF5 is not present in the normal human pancreas. Thus, we added this citation to our text and modified the statement in the Results section (3.1) as follows:
“We have previously shown that KLF5 is not present in the normal murine pancreas in pancreatic acinar cells [51]. Additionally, Diaferia et al., showed that KLF5 staining is absent in the normal human pancreas [55].”
Comment 4. There seem to be some redundant data. Either extra examples of the same data (e.g., 1B, supplementary 1A, etc.) can be further consolidated into summary data or they can be simply removed and/or described as further data not shown.
Response 4. We performed experiments using male and female mice, and in previous studies, the Reviewers were interested whether there is a difference in response to the disease based on gender. Therefore, we included data from female mice (main text) and male mice (supplementary results). We want to provide this information to the Readers, if feasible.
Comment 5. The presentation of amylase/lipase data is somewhat confusing. In cerulein-based rodent models of acute pancreatitis, theory suggests that these digestive enzymes are detected in the blood due to supraphysiologic hyperstimulation of the acinar cells that leads to aberrant basolateral secretion. Chronic pancreatitis has been modeled previously in rodents by three times weekly bouts of multiple cerulein for six weeks, with significant fibrosis being detected. In the present work, whereas some fibrotic changes are evident after only four weeks, the disease must be considered early, rather than late chronic CP as during late CP, due to accumulated acinar cell damage and fibrosis, loss of digestive function can be a factor and thus, the presence of robust serum levels of digestive enzymes following cerulein administration could no longer be assumed. As such, it is not clear which aspect of KLF5 deletion is responsible for the failure to obtain elevated blood levels of amylase and lipase in the KLF5-floxed mice with much less accumulated damage. The authors should try to reconcile this rhetorically, and/or substantiate KLF5-dependent reasons for the acinar cells to be unresponsive to the cerulein administration.
Response 5. We want to thank the Reviewer for this comment. The elevated levels of amylase/lipase result from the injury during acute pancreatitis. However, even in animal models of chronic pancreatitis that utilize cerulein to induce injury, the levels of amylase/lipase are not increased significantly after three/four weeks of injury (PMID: 30636940, 27095923, 23449669, 28496202). Additionally, we collected our samples three days after the final injection of cerulein to allow the resolution of acute changes. We added the following statement in the Discussion (section 4) “This could be because we collected samples for enzyme analysis three days after the final injection of cerulein or PBS and, thus, allowed the tissue to resolve the acute injury.”
In regard to minor comments, we made changes suggested by the Reviewer.
Minor comments.
Comment 1. L265, in the blood of mice, not “in the mice” should be emphasized.
Response 1. We replaced “in the mice” with “in the blood of mice”.
Comment 2. L450-1, “extensive” fibrosis in KRasG12D mice seems exaggerated to say at this early time point.
Response 2. We removed the word “extensive”.
Comment 3. L457, The sentence seems incomplete. “..treated”
Response 3. We completed the sentence as follows “with cerulein as compared to appropriate controls"
Comment 4. l.459, please use either “deletion of Klf5” OR “Klf5 inactivation” as using both is redundant.
Response 4. We removed “inactivation”.
Comment 5. l459, suggest, “Indeed” rather than “However”
Response 5. We replaced “However” with “Indeed”
Comment 6. l465, prefer “provide evidence that” to “demonstrate”
Response 6. We replaced “demonstrate” with “provided evidence that”
Comment 7. l481, promoters
Response 7. We modified it accordingly.
We hope that incorporated changes will satisfy the Reviewer and Editor and render the revised manuscript suitable for publication. Thank you for your consideration.
Sincerely,
Agnieszka Bialkowska, Ph.D.
Associate Professor
Renaissance School of Medicine at Stony Brook University
Department of Medicine
GI Translational Research Lab
HSC-T17 Room 090
Stony Brook, NY 11794-8176
Phone: (631) 638 2161
Email: [email protected]

Reviewer 2 Report
Comments and Suggestions for Authors
Alavi et al. report that KLF5 is overexpressed in chronic pancreatitis lesions in humans and mice and that deletion of Klf5 in the context of wild type or mutant Kras results in suppression of multiple types of events (fibrosis, metaplasia, inflammatory cytokine production, and inflammatory cell infiltration) related to damage preceding pancreatic cancer development. The work is well performed but represents an incremental step after their report from 2018 published in Gastroenterology. Overall, the work is interesting but not particularly innovative.
Specific comments
1. The authors only perform a single point analysis; a time-course would have been much more informative.
2. Figure 1. Even if this was published before, the staining of normal pancreas (organ donors) in parallel with the chronic pancreatitis lesions is desirable. The cores of some of the cases shown represent relatively preserved acinar structures and a comparison of the immunostaining with truly normal samples would be useful. The same applies to normal mouse pancreas.
3. The data shown in Figure 1C are very generic. The authors should provide a more detailed analysis of the different types of lesions (e.g. ADM, PanINs). Figure 1E is of insufficient quality. Higher magnification insets should be shown for all conditions.
4. Section 3.3: a more detailed quantification of the data presented in Figure 2A and accompanying suppl. material is necesssary.
5. Quantification of fibrosis, collagen production, etc. would be desirable for data related to Figure 3.
6. The authors assess the binding of KLF5 to the promoters of interest using 293 cells. This is useful but rather artificial. Since they have data on putative binding sequences, analysis by ChIP-qPCR of KLF5 binding to endogenous promoters in mouse pancreas in baseline and chronic pancreatitis conditions would be make the conclusions much stronger and physiologically relevant. In these experiments, the ChIP-qPCR of a control region should also be included.
7. The analysis of the inflammatory component shown in Figure 5 is very superficial and should be improved.
8. Similar data have been already reported for KLF4, therefore the authors should be addressed - ideally experimentally but otherwise in the Discussion - the different or overlapping roles of KLF4 and KLF5 in the various processes analyzed.
Minor comments
Please review carefully for grammar and typos. Some English editing is advisable.
Author Response
October 14th, 2022
Re: Cancer-1956515
We want to thank the Reviewer for his/her comments and suggestions. Please find our set-by-step response included below. Please notice that we separated Figure 1 into Figure 1 and Figure 2 and Supplemental Figure 1 into Supplemental Figures 1 and 2, and thus the numbers of figures have been modified accordingly.
Major questions.
Comment 1. The authors only perform a single point analysis; a time-course would have been much more informative.
Response 1. We agree with the Reviewer comment. We plan to perform time course studies using this model combined with inducible Klf5 deletion (DOX on/off system). However, these studies require at least another year to complete. At this stage, we would like to highlight the role of KLF5 in chronic pancreatitis after four months of treatment. In the animal model that we used, the four-week time point is well characterized and utilized by other researchers to demonstrate the relevance of different factors.
Comment 2. Figure 1. Even if this was published before, the normal pancreas (organ donors) staining in parallel with the chronic pancreatitis lesions is desirable. The cores of some of the cases shown represent relatively preserved acinar structures and a comparison of the immunostaining with truly normal samples would be useful. The same applies to normal mouse pancreas.
Response 2. We want to thank the Reviewer for this comment. Unfortunately, we were not able to obtain normal human pancreatic tissues. We provided the citation that provides the KLF5 stain in the normal human pancreas is, entitled "Dissection of transcriptional and cis-regulatory control of differentiation in human pancreatic cancer." Giuseppe R Diaferia, Chiara Balestrieri, Elena Prosperini, Paola Nicoli, Paola Spaggiari, Alessandro Zerbi, Gioacchino Natoli. The EMBO Journal (2016)35:595-617https://doi.org/10.15252/embj.201592404. The authors of this publication showed in Figure 5A that KLF5 is not present in the normal human pancreas. In addition, we provide an immunohistochemistry stain of KLF5 in Ptf1aCREERTM mice after tamoxifen and PBS treatment (Figure 2 and Supplemental Figure 2) that shows a lack of KLF5 stain in the murine pancreas.
Thus, we added this citation to our text and modified the statement in the Results section (3.1) as follows:
“We have previously shown that KLF5 is not present in the normal murine pancreas in pancreatic acinar cells [51]. Additionally, Diaferia et al. showed that KLF5 staining is absent in the normal human pancreas [55]."
Comment 3. The data shown in Figure 1C are very generic. The authors should provide a more detailed analysis of the different types of lesions (e.g. ADM, PanINs). Figure 1E is of insufficient quality. Higher magnification insets should be shown for all conditions.
Response 3. We want to thank the Reviewer for these observations. We provided in-detail characterization of the changes during chronic pancreatitis with the quantification of ADM, PanIN, edema, inflammatory infiltration, and loss of pancreatic ducts. The results for female mice are included in Figure 1, while the analysis for male mice is in Supplementary Figure 1. The IHC images for KLF5 (Figure 1E and Supplemental Figure 1D) have been moved to Figure 2, and Supplemental Figure 2, and we provided better quality images with magnification insets for all presented conditions.
Comment 4. Section 3.3: a more detailed quantification of the data presented in Figure 2A and accompanying suppl. material is necessary.
Response 4. We thank the Reviewer for this suggestion. We provided quantification KRT19/Amylase- positive acini in Figure 3 and Supplemental Figure 3. In addition, each figure includes quantification of Alcian Blue stain.
Comment 5. Quantification of fibrosis, collagen production, etc. would be desirable for data related to Figure 3.
Response 5. We provided data analysis for alpha-SMA staining and Masson's trichrome. These results are included in Figure 4 and Supplemental Figure 4. We did not provide quantification of Sirius Red images as the deletion of Klf5 almost completely inhibited fibrosis formation, as shown by Masson's trichrome and alpha-SMA stains. We hope that provided quantification will be sufficient.
Comment 6. The authors assess the binding of KLF5 to the promoters of interest using 293 cells. This is useful but rather artificial. Since they have data on putative binding sequences, analysis by ChIP-qPCR of KLF5 binding to endogenous promoters in mouse pancreas in baseline and chronic pancreatitis conditions would be make the conclusions much stronger and physiologically relevant. In these experiments, the ChIP-qPCR of a control region should also be included.
Response 6. We thank the Reviewer for this comment. We understand that endogenous studies would be beneficial. However, we do not have these results at this stage, and the experimental approach would require at least another eight months to perform these experiments. In our published work, we used an antibody against KLF5 that was previously tested for ChIP-qPCR. In addition, we used IgG as a negative control for all tested potential binding sites, which is a standard control for these types of experiments.
Comment 7. The analysis of the inflammatory component shown in Figure 5 is very superficial and should be improved.
Response 7. We agree with the Reviewer's comment. The deletion of Klf5 almost completely inhibited the number of CD45-positive cells. At this stage, we want to show that immune response is significantly inhibited upon Klf5 deletion, and we understand that we need to perform experiments in the future to address this topic.
Comment 8. Similar data have been already reported for KLF4, therefore the authors should be addressed - ideally experimentally but otherwise in the Discussion - the different or overlapping roles of KLF4 and KLF5 in the various processes analyzed.
Response 8. We thank the Reviewer for this comment. KLF4 role in pancreatic injury has been studied in a combination of KRAS G12D mutation in the setting of acute pancreatitis "KLF4 is essential for induction of cellular identity change and acinar-to-ductal reprogramming during early pancreatic carcinogenesis" Daoyan Wei, Liang Wang, Yongmin Yan, Zhiliang Jia, Mihai Gagea, Zhiwei Li, Xiangsheng Zuo, Xiangyu Kong, Suyun Huang, and Keping Xie. Cancer Cell. 2016 Mar 14; 29(3): 324–338. doi: 10.1016/j.ccell.2016.02.005. KLF4 and KLF5 have been shown to play an essential role in the development of ADM and mPanIN during acute injury. We provided the following statement in the discussion that hopefully addressed the Reviewer's comment and added appropriate references.
“KLF transcription factors have been previously shown to play an essential role in acute pancreatitis development and progression. Similarly to the inhibition of Klf5, the inactivation of Klf4 in the setting of the acute injury to the pancreas and the presence of KRASG12D mutation led to reduced levels of ADM and PanIN [66]. Conversely, Klf4 overexpression induced ADM and caused an increase in the ductal markers’ expression [66]. Furthermore, KLF4 and KLF5 have been shown to play a crucial role in pancreatic carcinogenesis and regulate processes such as proliferation and epithelial-to-mesenchymal transition [51,67-69]. These studies suggest that KLF4 can regulate ADM and PanIN formation processes during chronic pancreatitis, utilizing similar mechanisms as in acute injury.”
We hope the incorporated changes will satisfy the Reviewer and Editor and render the revised manuscript suitable for publication. Thank you for your consideration.
Sincerely,
Agnieszka Bialkowska, Ph.D.
Associate Professor
Renaissance School of Medicine at Stony Brook University
Department of Medicine
GI Translational Research Lab
HSC-T17 Room 090
Stony Brook, NY 11794-8176
Phone: (631) 638 2161
Email: [email protected]
Round 2
Reviewer 2 Report
Comments and Suggestions for Authors
The authors have addressed those issues of my previous comments that were easy to address, i.e. those that did not require any additional experiments. Critical aspects of the work such as checking different time points (which the authors misunderstood as I meant different time points of the pancreatitis rather than age of the mice), ChIP_PCR in pancreas or 266.6 cells, and others were dismissed. The statistical tests used (one-way anova) to assess the significance of the new quantifications is not appropriate for these comparisons, which are based on a small number of mice. Overall, none of the major issues raised has been addressed.